# Oral Vancomycin Prophylaxis for Primary and Secondary Prevention of *Clostridioides difficile* Infection in Patients Treated with Systemic Antibiotic Therapy: A Systematic Review, Meta-Analysis and Trial Sequential Analysis

**DOI:** 10.3390/antibiotics11020183

**Published:** 2022-01-30

**Authors:** Alberto Enrico Maraolo, Maria Mazzitelli, Emanuela Zappulo, Riccardo Scotto, Guido Granata, Roberto Andini, Emanuele Durante-Mangoni, Nicola Petrosillo, Ivan Gentile

**Affiliations:** 1First Division of Infectious Diseases, Cotugno Hospital, AORN Ospedali dei Colli, 80131 Naples, Italy; 2Infectious and Tropical Diseases Unit, Department of Medical and Surgical Sciences, Magna Graecia University, 88100 Catanzaro, Italy; m.mazzitelli88@gmail.com; 3Infectious and Tropical Diseases Unit, Padua University Hospital, 35121 Padua, Italy; 4Section of Infectious Diseases, Department of Clinical Medicine and Surgery, University of Naples “Federico II”, 80131 Naples, Italy; c.fid@hotmail.com (E.Z.); ri.scotto@gmail.com (R.S.); ivan.gentile@unina.it (I.G.); 5Clinical and Research Department for Infectious Diseases, National Institute for Infectious Diseases L. Spallanzani, IRCCS, 00149 Rome, Italy; guido.granata@inmi.it; 6Department of Precision Medicine, University of Campania ‘L. Vanvitelli’, 80138 Naples, Italy; roberto.andini@gmail.com (R.A.); emanuele.durante@unicampania.it (E.D.-M.); 7Unit of Infectious and Transplant Medicine, AORN Ospedali dei Colli-Monaldi Hospital, 80131 Naples, Italy; 8Infectious Diseases Service, Saint Camillus International University of Health Science, UniCamillus, 00131 Rome, Italy; nicola.petrosillo@unicamillus.org

**Keywords:** *Clostridioides difficile*, oral vancomycin, prevention, prophylaxis, meta-analysis, trial sequential analysis, antibiotic

## Abstract

Background: *Clostridioides difficile* infection (CDI) is associated with substantial morbidity and mortality as well as high propensity of recurrence. Systemic antibiotic therapy (SAT) represents the top inciting factor of CDI, both primary and recurrent (rCDI). Among the many strategies aimed to prevent CDI in high-risk subjects undergoing SAT, oral vancomycin prophylaxis (OVP) appears promising under a cost-effectiveness perspective. Methods: A systematic review with meta-analysis and trial sequential analysis (TSA) of studies assessing the efficacy and the safety of OVP to prevent primary CDI and rCDI in persons undergoing SAT was carried out. PubMed and EMBASE were searched until 30 September 2021. The protocol was pre-registered on PROSPERO (CRD42019145543). Results: Eleven studies met the inclusion criteria, only one being a randomized controlled trial (RCT). Overall, 929 subjects received OVP and 2011 represented the comparator group (no active prophylaxis). OVP exerted a strong protective effect for CDI occurrence: odds ratio 0.14, 95% confidence interval 0.04–0.38. Moderate heterogeneity was observed: I^2^ 54%. This effect was confirmed throughout several subgroup analyses, including prevention of primary CDI versus rCDI. TSA results pointed at the conclusive nature of the evidence. Results were robust to a variety of sensitivity and quantitative bias analyses, although the underlying evidence was deemed as low quality. No differences between the two groups were highlighted regarding the onset of vancomycin-resistant *Enterococcus* infections. Conclusions: OVP appears to be an efficacious option for prevention of CDI in high-risk subjects undergoing SAT. Nevertheless, additional data from RCTs are needed to establish OVP as good clinical practice and define optimal dosage and duration.

## 1. Introduction

*Clostridioides* (formerly known as *Clostridium*) *difficile* infection (CDI) is a leading cause of antibiotic-associated diarrhea (AAD), being implicated in 10–25% cases of AAD, as well as most cases of pseudomembranous colitis [1]. Although historically considered as a causative agent of healthcare-associated infections, *C. difficile* is increasingly prevalent in the community, causing disease even in subjects with no apparent risk factor [2].

CDI pathophysiology is driven by several factors. The pathogen can be acquired by multiple sources through person-to-person contact, exposure to contaminated environmental equipment and surfaces, or contact with temporarily colonized health care personnel [3]. Colonization is favored by the profound modification in the intestinal microbiota, that, in turn, may depend on either exogenous or host-related risk factors [4].

Among the former, pharmacological agents such as antibiotics and gastric acid suppressants (proton pump inhibitors, H2-receptor antagonists) stand out along with corticosteroids and nonsteroidal anti-inflammatory drugs; other exogenous determinants include invasive procedures and abdominal surgery [4]. Host-related factors include a high comorbidity burden, advanced age (over 65 years), immunosuppression (comprising neutropenia, bone marrow and solid organ transplantation), cancer, inflammatory bowel disease, chronic kidney disease and hypoalbuminemia [4].

This vast array of factors explains why the global burden of CDI has become so relevant worldwide in both hospital and community settings [2]. The major determinant of CDI remains the use of antimicrobials: A recent meta-analysis highlighted the strong association between CDI onset in hospital contexts and the use of carbapenems or third/fourth generation cephalosporins [5]. The role of antimicrobials is predominant also in community-acquired CDI, as demonstrated by a previous evidence synthesis [6].

CDI is characterized by a substantial mortality when presenting in either severe or fulminant form and, in general, among elderly and frail subjects [1]. Another challenge is represented by the high propensity of recurrence: After an initial, successfully treated episode, a recurrence occurs in 15–25% of patients within eight weeks [1]. A first recurrence increases the risk of further episodes, and the patient may enter a dramatic path made of multiple CDI recurrences [1]. The use of antibiotics after a previous episode as a driver of recurrent CDI (rCDI) is of utmost importance [7].

Several strategies have been attempted to prevent primary and rCDI. These can be divided into five major categories: active immunization, passive immunization, dysbiosis restoration, dysbiosis prevention and antibiotic prophylaxis [8]. The present work is aimed at evaluating, by systematically reviewing available data from the literature and through appropriate meta-analytic techniques including trial sequential analysis (TSA), the role of oral vancomycin prophylaxis (OVP) for primary and secondary prevention of CDI in subjects undergoing systemic antibiotic therapy (SAT).

## 2. Materials and Methods

The study protocol of this systematic review was prospectively registered to the International Prospective Register of Systematic Reviews (PROSPERO database, registration number: CRD42019145543) and was conducted according to the Preferred Reporting Items for Systematic Reviews and Meta-Analyses (PRISMA) guidelines, 2020 version (Appendix A) [9]. The protocol was initially registered in November 2019 and subsequently updated in order to extend the research and include further analyses (in particular, TSA).

### 2.1. Search Strategy

The first search was conducted from inception through 30 September 2019. The update implied a new search up to 30 September 2021. The following databases were screened: PubMed and EMBASE. The search was restricted to peer-reviewed articles. Neither language nor geographical restrictions were applied. The search strategies were conceived by two researchers of the team (A.E.M., M.M.), and adapted according to the databases’ features. Specific details are provided in Appendix A. Backward citation searching was conducted manually by inspecting the reference list of the papers eligible for full-text review in order to retrieve further articles.

### 2.2. Screening and Eligibility

Duplicate records were eliminated by using the EndNote 20 reference managing software (Clarivate, Philadelphia, PA, USA) [10].

The identified records were screened by titles and abstracts, as well as keywords. Papers with potential eligibility were then obtained for full text review. This two-step screening process for eligibility was performed by two authors (A.E.M., M.M.). Any disagreement was resolved through discussion within the entire study group.

Eligibility was firstly assessed by using the PICOS (population, intervention, comparators/controls, outcomes, study design) question format [11], as follows:Population: Patients undergoing SAT;Intervention: Administration of OVP;Comparators/controls: Standard of care, placebo, or any kind of intervention alternative to vancomycin;Outcomes: The main outcome of interest was the occurrence of a CDI episode, defined as first or recurrent; secondary outcome pertained to safety profile of intervention and was represented by the incidence of vancomycin-resistant *Enterococcus* species (VRE) infection;Study design: Randomized, non-randomized and observational (cohort, case-control, observational, cross-sectional) studies were included as long as they were peer-reviewed and published in full; case reports and series (fewer than 10 patients per group) were ineligible.

Studies that did not meet the aforementioned eligibility criteria were excluded.

### 2.3. Data Extraction

Data were extracted independently by two authors (A.E.M., M.M.) in a pre-defined Excel database. The following information was abstracted:Study-related variables (authors, year of publication, study country and design, setting);Patient-related variables (age, sex, underlying comorbidities);Infection-related variables (severity—severe versus mild-moderate forms, as defined by each study; epidemiological origin—community-onset versus healthcare-associated);Treatment variables (timing, sequence, dosage and duration of OVP and comparators);Concurrent SAT (type of drug, duration);Outcome variables (raw numbers of events and adjusted estimates if present).

The corresponding authors of included studies were contacted up to two times via e-mail requesting additional unpublished data.

### 2.4. Definitions

In accordance with international consensus, CDI was defined as onset of diarrhea, pseudomembranous colitis or toxic megacolon in a patient with a positive laboratory assay for *C. difficile* toxin A and/or B in stools or a toxin-producing *C. difficile* organism detected in stool via culture or other means (for instance, a positive molecular test) [12].

CDI cases with a positive *C. difficile* stool specimen between two to eight weeks of the last positive specimen were considered recurrent cases, provided that there was evidence of clear improvement of symptoms related to the previous episode, either after completion of initial treatment or within a week of CDI symptom onset if no treatment for CDI was administered [12].

The origin of CDI could be community-acquired, with onset of symptoms outside of healthcare facilities (and without discharge from a healthcare facility within the previous 12 weeks) or on the day of admission to a healthcare facility or on the following day in subjects not resident in a healthcare facility within the previous 12 weeks, or healthcare-associated. In the latter instance, the onset of symptoms was on day three or later following admission to a healthcare facility on day one, or within four weeks of discharge from a healthcare facility (including the current hospital or a previous stay in any other healthcare facility) [12].

Primary prophylaxis was defined as prevention of an episode of CDI in a subject with no previous history of CDI, whereas secondary prophylaxis was the prevention of CDI recurrence.

For the purpose of the present study, even different definitions adopted from primary studies of the abovementioned events were taken into account and critically interpreted.

### 2.5. Data Synthesis and Analysis

Anticipating a meta-analysis of rare events, this work was conducted within a proposed framework to guide evidence synthesis in case of zero-events studies [13].

Pooled odds ratios (OR) and their 95% confidence intervals (CIs) were calculated by using a random effects model via generalized linear (mixed-effects) model (GLMM) with exact noncentral hypergeometric-normal likelihood (NCHGN) [14], a one-stage approach in which each study is considered as a stratum or cluster, and the overall effect size can be computed by the population average method [13]. Risk difference (RD) was used as an alternative to OR when appropriate according to the aforementioned framework [13].

Statistical heterogeneity among studies was assessed by the Cochran’s Q statistic and the I^2^ statistic (degree of heterogeneity): the heterogeneity was considered low, moderate or high for I^2^ values of less than 50%, 50% to 74%, and 75% or greater, respectively [15].

Together with the pooled effect sizes of binary outcomes and the 95% CIs thereof, a prediction interval (PI) was reported, reflecting the variation in treatment effects over different settings, including what effect is to be anticipated in future patients [16].

To better convey and interpret findings, ORs were expressed as number needed to treat (NTT) across a range of assumed comparator risks (ACRs). NNT was computed from OR by means of proper formulas and was defined as the expected number of subjects who need to receive the experimental rather than the comparator intervention for one additional individual to either experience or avoid an event (depending on the direction of the result) in a given time frame [17].

Subgroup analyses based on categories defined a priori were conducted as well. The most important subgroups involved primary and secondary prophylaxis. Other subgroups were the following: study location, study design, type of population, OVP dose, mean duration of OVP (longer or shorter than mean concurrent SAT), timing of follow-up and number of previous episodes of CDI for secondary prophylaxis (history of one episode versus at least two).

If present, adjusted treatment effects from studies that included a multivariable analysis of CDI occurrence were pooled and meta-analyzed by resorting to the inverse variance method. The adjusted OR (aOR) was the effect size of choice to pool adjusted data. If an aOR was not present, adjusted risk ratio (aRR) or hazard ratio (aHR) were conveniently converted to aOR. If adjusted effect size with its 95% CI was not available but OVP was reported not to have a significant impact on the primary outcome, an aOR of 1 was imputed, and the standard error (SE) of the unadjusted analysis was used as the measure of dispersion, as described elsewhere [18].

Furthermore, meta-regression analysis was planned to explore study-level sources of heterogeneity for continuous moderators on the dependent variable of primary interest, namely CDI occurrence through the maximum-likelihood estimation method. Although the minimum number of studies per covariate in meta-regression analyses required to minimize the risk of overfitting is not established, a minimum of 10 studies per examined covariate was taken into account [19]. As for continuous variables (e.g., days of therapy), means and standard deviations (SD) if not provided were derived from sample size, median, interquartile range (IQR) and minimum and maximum values [20].

In addition, a TSA was performed to assess whether the results regarding the primary outcome (effect of OVP on CDI prevention) would be subject to type I or type II errors associated with sparse data and a lack of power. TSA combines traditional meta-analysis methodology with repeating significance testing methods applied to accruing data in clinical studies [21]. Before the required information size (RIS) is reached, TSA constructs monitoring boundaries to establish when an estimated effect is so convincingly large (or small) that the conclusions are unlikely to change with additional evidence. A model variance-based diversity-adjusted information size was used for the TSA based on α = 0.01 and β = 0.10 (power of 90%) to be more conservative, and on a clinically relative risk reduction (RRR) of 70%. The conservative trial sequential monitoring boundaries were set by O’Brien–Fleming as the α spending function. The cumulative Z-curve of each cumulative meta-analysis was computed and plotted against the above monitoring boundaries. The crossing of the cumulative Z-curve into the trial sequential monitoring boundary for benefit points to the fact that a sufficient level of evidence has been reached, and no additional studies may be needed to demonstrate the superiority of the intervention. If the cumulative Z-curve does not cross any of the trial sequential monitoring boundaries, there is probably not sufficient evidence to reach a conclusion and further studies may be needed. If the cumulative Z-score curve crosses into the futility area boundary, future trials are unlikely to modify the trend of evidence. OR was used as the effect measure for meta-analysis in the context of TSA, and effect size was integrated by means of the Biggerstaff–Tweedie (BT) method (a random effects model) that incorporates the uncertainty of estimating the between-trial variance and minimizes the effect of the bias especially when the size of the trials varies [22].

Eventually, several sensitivity analyses were conducted concerning the primary outcome. An alternative method to generate the pooled effect size was used, in particular a Mantel–Haenszel (MH) method without continuity correction, a two-stage approach (known as the standard meta-analysis approach as well) in which the study-specific effect sizes and standard errors are obtained or estimated from the available studies at the first stage, and subsequently they are synthesized at the second stage [13]. For quantitative bias analyses to gauge the robustness of the results, the E-value was calculated, namely the minimum strength of association that an unmeasured confounder would need to have with both the treatment and the outcome to fully explain away a specific association; in this case, with the occurrence of CDI [23]. An evaluation of the impact of potential outliers was run. Indeed, the contribution of each study to the observed heterogeneity was graphically evaluated through a Baujat plot [24]. A leave-one-out sensitivity analysis was carried out to assess the influence of individual studies, primarily the ones contributing to heterogeneity, on the summary estimate. Additionally, a Graphical Display of Study Heterogeneity (GOSH) plot (x-axis = summary log OR, y-axis = I^2^) was generated by fitting the same meta-analysis model to one million randomly sampled subsets of the studies to identify subclusters with different effect sizes [25]. Studies most contributing to subclusters were identified with clustering algorithms and classified as potential outliers. As further sensitivity analysis, a meta-regression by means of an alternative method (MH without continuity correction) was performed.

All calculations for the meta-analysis were performed by R statistical software version 4.0.3 (R Project for Statistical Computing, Vienna, Austria) in RStudio statistical software version 1.3.1093 with the following packages: ‘meta’, ‘metafor’, ‘dmetar’. TSA was conducted using Trial Sequential Analysis software version 0.9.5.10 beta (Copenhagen Trial Unit, Centre for Clinical Intervention, Copenhagen, Denmark). All *p* values < 0.05 were considered statistically significant.

### 2.6. Publication Bias and Quality Appraisal

Publication bias and small-study effects were assessed by visual inspection of contour-enhanced funnel plots [26] and the regression test by Egger [27]. The Methodological Index for Non-Randomized Studies (MINORS) was used for the risk of bias assessment of the included studies when observational in nature [28]. The score assesses 12 items and a maximum score of 24 can be achieved. Studies with a score lower than 16 were deemed at high risk for bias, low if the score was at least 20, medium for intermediate values (16–19). With regard to randomized controlled trials (RCT), the revised tool to assess risk of bias in randomized trials (RoB 2) was implemented [29] by resorting to the *robvis* web app [30]. The Grading of Recommendations Assessment, Development and Evaluation (GRADE) tool, by means of the GRADEpro Guideline Development Tool webpage, was used for a global evaluation of the body of evidence in the systematic review in order to gauge the certainty (quality) of the findings as confidence in the estimate (magnitude) of effect [31]. Initial level of confidence relies on study design (high for RCT, low for observational studies), whereas reasons for rating down confidence are based on the following elements: risk of bias, imprecision, inconsistency, indirectness and publication bias. On the contrary, especially in case of observational studies, rating up the quality of evidence is possible if one of the subsequent criteria is met: (i) when a large magnitude of effect exists; (ii) when there is a dose-response gradient; (iii) when all plausible potential confounders or other biases augment the confidence in the estimated effect [32].

## 3. Results

### 3.1. Study Selection

The most updated study search yielded 978 records. After de-duplication, 664 studies (including one subsequently retrieved by handsearching of references) were selected for screening by titles and abstracts. In the following step, 32 studies were considered eligible for full-text review, and eventually 11 fulfilled the inclusion criteria [33,34,35,36,37,38,39,40,41,42,43]. The study inclusion flow diagram is described in Figure 1.

### 3.2. Study Characteristics

Details of included studies are available in Table 1. Only one study was a RCT [42], with all the others being retrospective observational studies [33,34,35,36,37,38,39,40,41,43]. Publication year ranged from 2016 to 2021, with study periods from 2010 to 2019. Most studies (9 of 11) were conducted in the United States (US), while one was performed in Canada [33] and another one in Croatia [35].

Overall, 2940 subjects were included, 929 in the OVP group and the remaining 2011 in the comparator group, composed only of individuals not receiving any potentially active prophylaxis for CDI (henceforward defined as group of either no intervention or no active prophylaxis). All but one study [43] involved adult subjects, unselected (largely hospitalized) or with specific baseline conditions such as solid organ transplantation (SOT) [36,37] or hematological malignancies/recent history of hematopoietic stem cell transplantation (HSCT) [39,41]. In all instances, subjects were administered SAT, whose type and duration differed widely; as for the latter, only in one case mean time of antibiotic exposure was less than seven days [38], with median values ranging from 10 to 14 days.

Four studies addressed primary prophylaxis [35,37,39,42], while seven addressed secondary prophylaxis of CDI though OVP [33,34,36,38,40,41,43]. Of note, in one study, mainly focused on primary prophylaxis, 15.6% of patients in the OVP arm had a history of CDI, compared to 9.1% in the comparator group; since data could not be split and the majority of data regarded primary prevention, the study was considered in the homonymous subgroup [39].

Timing of follow-up varied across studies, from in-hospital to 12 months after the first CDI episode [33]. The definition of the main event informing the primary outcome was not perfectly uniform, but generally depended upon the combination of diarrhea plus a positive molecular test.

OVP dosage varied: In most instances, it was 125 mg once-daily (OD) [35,42] or bis in die (BID) [36,37,39,41,43].

In some studies, no primary events occurred in only one arm [35,36,39,42]; according to the proposal by Xu and colleagues, the meta-analysis including these studies was classified as ‘MA-SZ’: evidence synthesis containing zero-events only occurring in single arms, no double-arm-zero-events studies are included, and the total events count in neither arm is zero [13].

Less than 50% of studies reported adjusted data (4 out of 11) [33,38,41,43].

Only five studies reported VRE infections as a safety outcome (for a total of 996 subjects): two studies focused only on bloodstream infections [37,39], the other three on VRE infections overall [40,41,43] during the same follow-up time dedicated to the primary outcome. Always referring to the framework set by Xu and colleagues for meta-analysis with zero-event studies [13], the evidence synthesis concerning the risk of VRE infections under OVP for CDI was classified as ‘MA-DZ’: meta-analysis contains zero-events only occurring in double arms, and the total events count in neither arm is zero.

### 3.3. Primary Outcome

The CDI proportion (incidence) was 11.2% (104/929) in the OVP group and 14.9% (300/2011) in the arm without active prophylaxis. The use of OVP was associated with a lower CDI occurrence compared with no intervention (OR 0.13, 95% CI 0.04–0.38), as shown in Figure 2. Moderate heterogeneity was detected (I^2^ = 54%). The PI ranged from 0.01 to 3.06. The forest plot in Figure 2 depicts results regarding the main subgroups as well, namely primary and secondary prophylaxis. The treatment effect was stronger in the first subgroup (OR 0.03; 95% CI 0.00–0.18, I^2^ = 0%) than in the second one (OR 0.31, 95% CI 0.13–0.71, I^2^ = 71%); notably, the results of the subgroup analysis suggested that there was a statistically significant subgroup effect (*p* = 0.02).

Overall, the NNT was six (95% CI 5–9), meaning that six subjects under SAT need OVP rather than no intervention to prevent one additional CDI; this applies to a scenario of an ACR equal to 19%, the mean value extrapolated from the dataset. For an ACR of 6% (minimum value), the NNT was 19 (95% CI 18–28); for an ACR of 34% (maximum value), the NNT was four (95% CI 3–6).

Adjusted effect size from studies reporting multivariable analyses (all regarding secondary prophylaxis) [33,38,41,43] were pooled along with an imputed adjusted effect size equal to one (with standard errors borrowed from unadjusted analysis) as to studies in which the intervention did not impact significantly on the outcome under investigation [36,37,42]. The overall result was an aOR of 0.52 (95% CI 0.38–0.70, I^2^ = 0%), with a narrow PI ranging from 0.34–0.77. As depicted in Figure 3, the result was consistent among subgroups of primary and secondary prophylaxis. In Appendix A, the variables used in the different studies to provide an adjusted estimate between the exposure (OVP) and the outcome (CDI onset) are reported.

Several a priori subgroup analyses were conducted in the overall population under investigation (Table 2). In all cases OVP confirmed its protective effect on CDI onset, nevertheless the treatment effect, though consistent as direction was concerned, differed in magnitude. For instance, it was larger (with narrow CI) in the following subgroups: patients affected by hematological disorders (OR 0.03, 95% CI 0.00−0.23, I^2^ 0%) and patients with in-hospital follow-up (OR 0.03, 95% CI 0.00−0.22, I^2^ 0%). In contrast, the treatment effect was smaller in subjects with 90-day follow-up (OR 0.82, 95% CI 0.61−1.11, I^2^ 0%), in individuals whose mean OVP duration in days was inferior to mean SAT duration (OR 0.44, 95% CI 0.16−1.23, I^2^ 40.1%) and, eventually, in persons receiving OVP dose different from 125 mg once daily or twice a day (usually higher with at least 500 mg/day), as shown by an OR equal to 0.43 (95% CI 0.15−1.23, I^2^ 78.5%). Test for subgroup difference was significant only for follow-up time span and mean duration of OVP (Table 2).

Another subgroup analysis involved specifically the use of oral vancomycin as secondary prophylaxis measure, stratifying patients according to the history of prior episodes. Useful data came from all studies addressing secondary prophylaxis but one [34]. The related forest plot is reported as Appendix A. OVP proved its effectiveness in both subgroups, with no significant test for interaction: OR 0.34 (95% CI 0.13–0.89, I^2^ 60%) in the subgroup having a history of one prior CDI episode, OR 0.33 (95% CI 0.10–1.06, I^2^ 75%) in the other subgroup (at least two past episodes).

Eventually, a meta-regression analysis was carried out to study the impact of OVP duration (days) on the primary outcome (CDI occurrence, whichever the context, either primary or secondary prophylaxis), as shown in Appendix A. Random effects meta-regression revealed a statistically significant association between the log OR for CDI onset and the mean duration of OVP (*p* < 0.001). The slope coefficient, reflecting the change in the average log OR for a one-unit (i.e., days of OVP) increase, was −0.18 (95% CI from −0.28 to −0.07), with a standard error equal to 0.05. Adjusted R^2^, which is used to quantify the proportion of variance explained by the covariates, was not computed by the GLMM meta-regression.

#### 3.3.1. Primary Outcome: Trial Sequential Analysis

In our trial sequential analysis, the type I error risk was set at α = 0.01 with a power of 0.90 and an anticipated RRR linked with intervention equal to 70%. Under these premises, the RIS for the meta-analyzed estimate was 2114 (Figure 4). Thus, TSA confirmed the results obtained in the conventional meta-analysis had reached the RIS, with the number of included subjects equal to 2940. The Z-score curve (blue line) crossed both the required information size (vertical red line) and the conventional statistical significance boundary corresponding to two-sided *p* value of 0.05 (horizontal red lines), indicating that the observed reduction in rate of CDI in subjects taking OVP could be considered conclusive with the existing evidence. Diversity (D^2^), defined as the proportion of the total variance in a random-effects model contributed by the between trial variation despite its estimator, was 88%.

#### 3.3.2. Primary Outcome: Sensitivity Analyses

To further assess the robustness of our results, a meta-analysis through another appropriate method for a work classified as MA-SZ was first carried out. In particular, a MH method was used. The results were consistent: The use of OVP was protective towards CDI onset when compared with no intervention (OR 0.22, 95% CI 0.10–0.49), as depicted in Appendix A. Higher heterogeneity was detected (I^2^ = 70%), but the PI was narrower (from 0.02 to 2.46) compared to the main analysis. At variance with the latter, the test for subgroup difference (between primary and secondary prevention groups) was not statistically significant.

Sensitivity analysis using E-value suggested that the observed association between OVP and CDI prevention was quite robust to potential unmeasured confounding. For instance, to explain an OR of 0.13 for CDI occurrence, an unmeasured confounder associated with both increased likelihood of OVP and a decreased likelihood of CDI by risk ratio of 14.87-fold above the measured covariates could suffice, but weaker confounding could not (Appendix A). To move the upper CI to include the null value, an unmeasured confounder associated with increased OVP use and decreased CDI by risk ratio of 4.7-fold could suffice, but weaker confounding could not.

Moreover, the robustness of results was tested against the presence of outliers and influential cases. Traditional leave-one-out analysis demonstrated that the positive effect of OVP on CDI occurrence was not overturned by removing one specific study but was slightly attenuated when omitting the study by Ganetsky and colleagues [39]: OR 0.18 (95% CI 0.07–0.48, I^2^ 59%). Slightly larger effect size was observed if omitting Carignan et al. (OR 0.10, 95% CI 0.03–0.30, I^2^ 53%) [33] or Caroff et al. (OR 0.10, 95% CI 0.03–0.30, I^2^ 52%) studies [38]. Of note, heterogeneity fell from moderate to low (I^2^ 35%) when not taking into account the study by Van Hise and collaborators [34] (Appendix A).

To more closely explore this aspect, a Baujat plot was constructed, depicting the overall heterogeneity contribution for each study against the influence on the pooled result for each study (Appendix A). The latter study confirmed to be the one providing the highest overall heterogeneity [34], whereas the highest influence by far on pooled results was exerted by the Canadian study from Carignan and colleagues [33]. GOSH analysis highlighted the following pattern in the data: While most values were concentrated in a cluster with relatively high effects and moderate to high heterogeneity, the distribution of I^2^ values was heavily right-skewed and bi-modal (Appendix A). The ensuing diagnostics confirmed the influential role by the three aforementioned studies (Carignan, Caroff and Ganetsky) [33,38,39]. Re-running the meta-analysis while removing the three studies had a large impact on the estimated heterogeneity (I^2^), dropping it to zero (Appendix A). The effect size essentially overlapped with that of the main analysis: OR 0.10, 95% CI 0.05–0.22. Besides the narrower CI, the exclusion of these outliers led to an overall PI not containing the null or opposite effect (from 0.04 to 0.26). The test for subgroup difference was not statistically significant (*p* = 0.43).

Finally, a sensitivity analysis was performed on the meta-regression investigating the relationship between OVP duration (days) and the main outcome. Running the meta-regression in the context of a MH method without continuity correction, a statistically significant impact of the covariate on CDI onset was confirmed (*p* = 0.0017). The slope coefficient was −0.12 (95% CI from −0.20 to −0.05), with a standard error equal to 0.04 (Appendix A). Adjusted R^2^, representing the amount of heterogeneity, accounted for was 77%, which means that 77% of the difference in true effect sizes can be explained by the duration of OVP.

### 3.4. Secondary Outcome

The proportion (incidence) of VRE infections was 2.4% (6/255) in the OVP group and 0.9% (7/741) in the comparator group. The use of OVP was not associated with a higher risk of VRE infections as a complication (RD equal to zero, 95% CI −0.02 to 0.01), as illustrated in Figure 5. No heterogeneity was detected (I^2^ = 0%). The PI ranged from −0.03 to 0.02. No statistically significant subgroup effect was detected (*p* = 0.65).

### 3.5. Publication Bias and Quality Appraisal

In Appendix A, the contour-enhance funnel plot depicts small-study effects as proxy of publication bias, showing how asymmetry patterns relate to statistical significance. The funnel plot is asymmetrical, showing that three small studies all had significant effects, despite having a large standard error. The two studies with the smallest standard error instead are distributed more symmetrically on the plot. Beyond inspection, asymmetry in the funnel plot was confirmed quantitively by Egger’s regression test, whose result was statistically significant (*p* = 0.004).

The risk of bias was assessed by resorting to different tools according to the study design, whether interventional (RCT) or observational. In Appendix A, traffic light plot about the evaluation of the only RCT included [42] is reported; the study was judged as not at low risk of bias, raising some concerns especially regarding unclear randomization process and missing outcome data. All observational studies were considered at a moderate risk of bias (Appendix A).

Certainty of obtained evidence was judged in the context of the GRADE approach as illustrated in Appendix A. As far as the primary outcome was concerned, the certainty of evidence in favor of OVP was deemed low: elements to rate down were the large prevalence of observational studies not free of potential biases, and their relative inconsistency; factors to rate up were large effect size and potential dose-response gradient based on increasing effect with growing OVP duration. The certainty of evidence for the secondary outcome (no difference between OVP use and no use of OVP in terms of VRE infections) was considered very low.

## 4. Discussion

To the best of our knowledge, this is the first systematic review and meta-analysis with trial sequential analysis addressing the clinical question of the effectiveness of OVP in CDI prevention in persons undergoing SAT. Two earlier meta-analyses have been published on the topic [44,45] that informed a recent guideline endorsed by the American College of Gastroenterology (ACG) [46]. ACG experts conditionally recommend OVP for secondary CDI prevention in high-risk patients undergoing SAT, although the recommendation is supported by low-quality evidence [46]. The ACG panel defines high-risk subjects as those aged 65 years or older or with significant immune system alterations who were hospitalized for severe CDI within the past three months [46]. Moreover, ACG guidelines suggest using low-dose vancomycin, i.e., 125 mg once daily, to be stopped five days after completion of SAT [46]. The latest guidelines by the Infectious Disease Society of America only focus on the use of fidaxomicin and bezlotoxumab for the treatment of CDI [47], whereas the 2021 guidelines by the European Society of Clinical Microbiology and Infectious Diseases (ESCMID) specifically address the issue of CDI prevention (clinical question number nine) [48]. The ESCMID committee does not support the routine use of anti-CDI antibiotics during SAT (good practice statement), leaving a window open for very select patients with a history of multiple rCDI incited by SAT, always after careful balancing of the risks and benefits, preferably with the support of an infectious disease or a clinical microbiology specialist [48].

There are key differences between the present work and the two previous meta-analyses [44,45]. Babar and colleagues included fewer studies and, performing several meta-regression analyses, did not investigate the impact of OVP duration [44]. On the other hand, among the many variables tested, they showed that higher doses of OVP may magnify the risk of CDI, also accounting for 100% of heterogeneity among studies [44]; this finding is coherent with results of our work, since from subgroup analysis the lowest OVP dosage (125 mg once daily) emerged as the best. Tariq and collaborators included 14 studies, but one-third (5 out of 14) were conference presentations [45]. In the former meta-analysis, the OR for CDI onset associated with OVP use was 0.26 (95% CI 0.13–0.52), in the latter 0.34 (95% CI 0.20–0.59). These data were the foundation for our choice to set a 70% RRR as prerequisite for the TSA.

In our study, the protective effect of OVP was even more evident. Vancomycin has potent and well-known activity against *C. difficile*, does not have the long-term systemic adverse events associated with other drugs such as metronidazole, and has an inferior cost compared with fidaxomicin, appearing as a logical choice for CDI prevention [49]. Of note, both metronidazole and fidaxomicin have been tested in randomized studies for primary prevention of CDI: metronidazole use did not result in a reduction in antibiotic-associated diarrhea [50], and fidaxomicin failed to meet the main composite outcome in a dedicated RCT in HSCT adults, although significantly reducing CDI episodes compared with placebo [51]. The issue with vancomycin lies in the largely unknown downstream effects of its long-term use or recurrent exposure, especially when it comes to development of resistance and promotion of VRE colonization [52]. Indeed, as early as the 1990s, a paradoxical effect of vancomycin was observed: An increased risk of CDI compared with placebo at the end of a two-month follow-up after the completion of an antibiotic course aimed at eradicating asymptomatic *C. difficile* colonization [53].

In experimental animal models, vancomycin showed a profound effect on the microbiome, through which a loss of colonization resistance to VRE, other multidrug-resistant organisms (Gram negative rods in particular) and *C. difficile* itself occurs [54]. In humans, moreover, elevated oral vancomycin doses (>500 mg/day) have been associated with subsequent development of bloodstream infections due to *Candida* spp. and Gram-negative bacteria, owing to a likely mechanism of microbial translocation because of drug-induced intestinal tissue damage [55]. The results of our meta-analysis are consistent with this finding, since the lowest dose of OVP appeared to exert the maximum benefit, probably minimizing the adverse events.

Another issue is represented by the increasing risk of resistance to vancomycin by *C. difficile*, a phenomenon that has become clinically and statistically relevant in the last decade [56]. Therefore, physicians need to cope with a double-edged sword in that too high OVP doses may abolish the CDI prevention effects of vancomycin by increasing dysbiosis, while too low doses may enhance the likelihood of resistance development [52].

At the same time, subgroup analysis showed that effectiveness of OVP was more pronounced when its duration was longer than concurrent SAT. Moreover, according to our meta-regression, the longer the OVP duration, the stronger the effect; this is in line with ACG recommendations upon discontinuation of OVP five days after completion of SAT [46]. Whether OVP should be stopped when SAT ends or should be prolonged, and in this case for how long, remains an open issue [57].

Notwithstanding the lack of head-to-head studies, OVP appears very appealing as opposed to other potential preventive strategies for CDI, such as fecal microbiota transplantation (strategy of dysbiosis restoration) [58] or passive immunization with bezlotoxumab [59], in the light of its very low cost and ease of administration [60].

The results of our meta-analysis underscore the usefulness of OVP in persons at high risk, with this finding being consistent across several subgroups, confirmed by pooling adjusted effect sizes and by TSA, and robust to numerous sensitivity and quantitative bias analyses. For instance, the average effect calculated was not too heavily biased by outliers and influential studies, since the OR remained similar when omitting the following studies stemming from GOSH diagnostics: Carignan [33] and Ganetsky [39], whose narrow CI suggests high weight, as well as Caroff [38], characterized by the largest sample size (760) [38].

Nonetheless, these results are generated from mostly observational studies not free of biases: The measure of the related inherent uncertainty is represented by the PI from the main analysis including the opposite effect, although that does not happen in the adjusted analysis and in the analysis omitting the most influential studies. Therefore, pending the results of recently completed (NCT03200093) and ongoing RCTs (NCT02996487, NCT04000555, NCT03462459, NCT03466502) that will provide high-quality evidence, we believe OVP should be offered as a well-tailored strategy by accurately selecting subjects that could benefit from it.

Risk factors for primary [61] or rCDI [62] are well known. There are several prediction models for rCDI [63] and others for primary infections [64]. The likely future, against the backdrop of precision medicine, would be to exploit big data analysis and machine learning approaches in order to make accurate predictions by identifying in advance subjects on the verge of developing CDI, especially in a hospital setting [65]. They may be the focus of targeted interventions, such as withholding or reducing the exposure to high-risk antibiotics or, if not possible, introducing OVP.

The strengths of our work include a comprehensive literature search with rigorous inclusion criteria, excluding works not published in full. Furthermore, as opposed to previous meta-analyses on the topic [44,45], we followed a detailed statistical framework to deal with zero-event studies [13] and performed a TSA, a powerful tool to assess the conclusiveness of meta-analyses [66]. Furthermore, we conducted several sensitivity and quantitative bias analyses, eventually carefully assessing the available evidence according to the GRADE system.

Of course, this study presents some limitations. The large majority of included studies were observational in nature, limiting the generalizability of results. Although an E- value of 14.87 suggested that a confounder, or set of confounders, would have to be associated with a nearly 15-fold increase in the risk of CDI and must be almost 15 times more prevalent in subjects undergoing OVP than in persons not under prophylaxis to explain the observed effect size, the role of a lurking variable cannot be excluded, since the obvious bias in these studies is represented by the unknown factors that dictated prescribing OVP. The other relevant issue is constituted by the great heterogeneity among studies as far as key aspects are concerned: (i) type and duration of SAT; (ii) the role of factors other than antimicrobials influencing CDI risk; (iii) dosage and duration of OVP; (iv) the definition of the CDI episode, not always consistent with well-established guidelines (sometimes defined only by positive *C. difficile* testing with or without diarrhea, making it difficult to differentiate from colonization and real infection); (v) the high variability in follow-up, especially for rCDI, sometimes going beyond the eight weeks described by international guidelines, such that Tariq and colleagues prefer the expression ‘future CDI’ [45]; (vi) the huge difference in timing of prior CDI in the context of secondary prevention as well. Granular data to assess were not always available to explore all the variables of interest. For instance, the difference in SAT could not be disentangled. However, in the majority of cases, the most used concurrent antibiotic belonged to a ‘high risk’ class such as cephalosporins [63,67]. Similarly, the role of concomitant inciting factors for CDI including, among the many, acid-suppressing medications or steroids, could not be addressed. For instance, the data on the concurrent use of gastric acid suppressant par excellence such as pump proton inhibitors (PPIs) were too sparse to assess the potential interaction of OVP with PPIs on CDI occurrence. At any rate, considering the well-known role of PPIs as risk factor for CDI [6], a so-called stewardship of acid-suppressing medication is warranted through systematic de-prescribing [68] Another limitation concerns the safety profile of OVP, because the rate of VRE infections is not the best measure to assess its potential negative impact on microbiota. Instead, the rate of VRE colonization should be evaluated. Eventually, the cost-effectiveness of OVP could not be evaluated in depth, especially in the face of the potentially high cost of generic vancomycin capsule, usually the formulation available for outpatients, when compared with oral solution from vials (originally intended for intravenous infusion) available in health-care settings [69].

## 5. Conclusions

OVP represents a promising preventive weapon for both primary and recurrent CDI, and authoritative guidelines have already endorsed it, although cautiously in the light of the low-quality underlying evidence. Many lingering questions remain about its exact schedule (dose and duration) as well as the ideal patient profile benefitting the most from this approach. Pending further data, a prudent strategy would be represented by the use of OVP at a low dose (125 mg once or twice daily) in very selected subjects, namely the ones undergoing SAT with high-risk antibiotics and having relevant likelihood to develop CDI according to available prediction models. Well-conducted RCTs will shed some light on the aspects still in search of an answer regarding this preventive strategy.

## 6. Ethics

No ethics committee approval or subject informed consent was necessary since deidentified patients’ data of already published studies were analyzed.

## Figures and Tables

**Figure 1 antibiotics-11-00183-f001:**
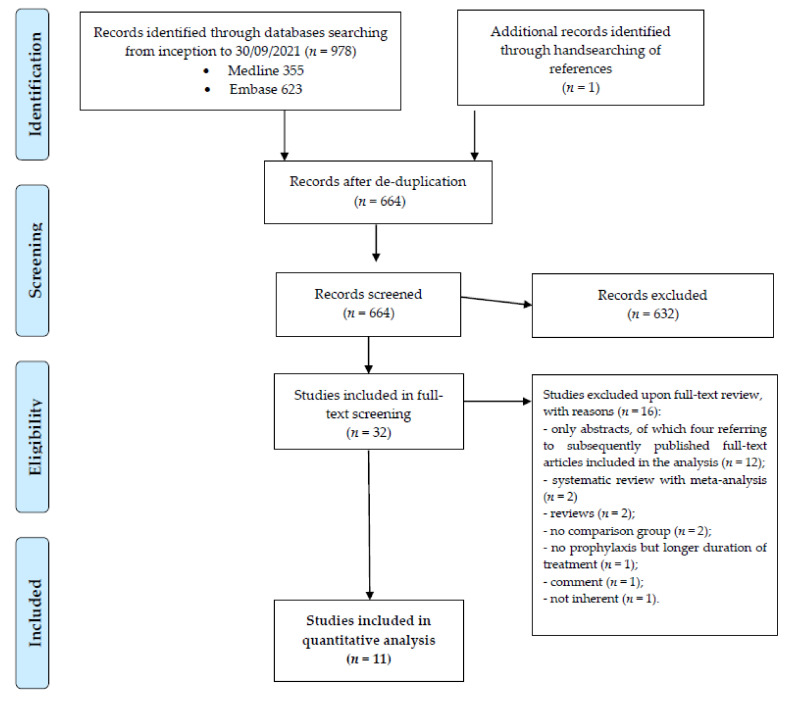
Results of literature search and flow diagram for selection of eligible studies.

**Figure 2 antibiotics-11-00183-f002:**
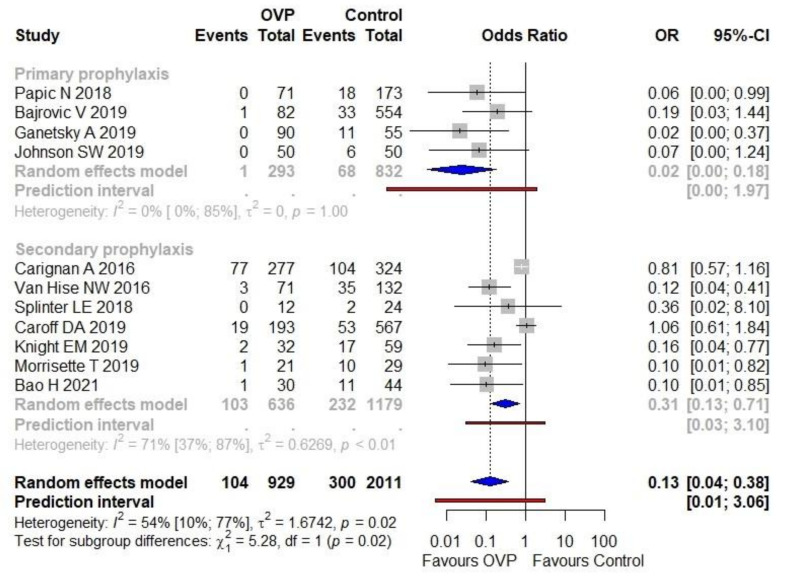
Meta-analysis regarding the association of OVP with CDI prevention, overall and across the principal subgroups. Abbreviations: CDI, *Clostridioides difficile* infection; OR, odds ratio; OVP: oral vancomycin prophylaxis; 95%-CI, confidence intervals at 95%. Vertical line indicates ‘no difference’ point between the two options. Squares represent odds ratios. Diamonds represent pooled odds ratios for all studies. Horizontal lines represent 95% CI.

**Figure 3 antibiotics-11-00183-f003:**
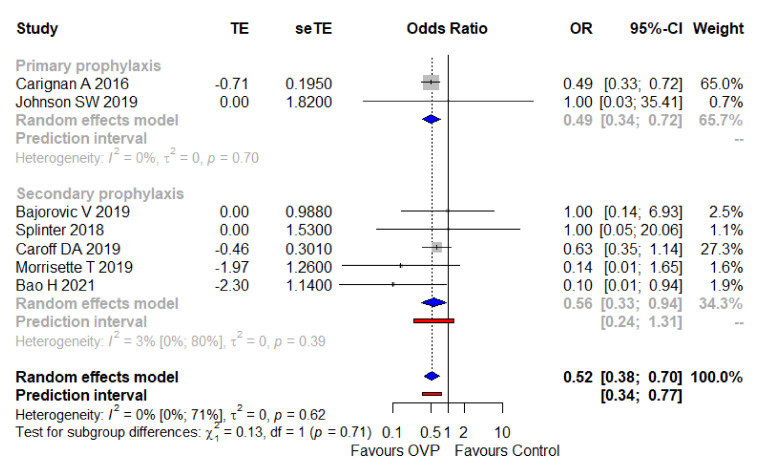
Meta-analysis regarding the adjusted odds ratio of CDI with OVP, overall and across the main subgroups. Abbreviations: CDI, *Clostridioides difficile* infection; OR, odds ratio; OVP, oral vancomycin prophylaxis; seTE, standard error treatment effect; TE, treatment effect; 95%-CI, confidence intervals at 95%. Vertical line indicates ‘no difference’ point between the two options. Squares represent adjusted odds ratios. Diamonds represent pooled adjusted odds ratios for all studies. Horizontal lines represent 95% CI.

**Figure 4 antibiotics-11-00183-f004:**
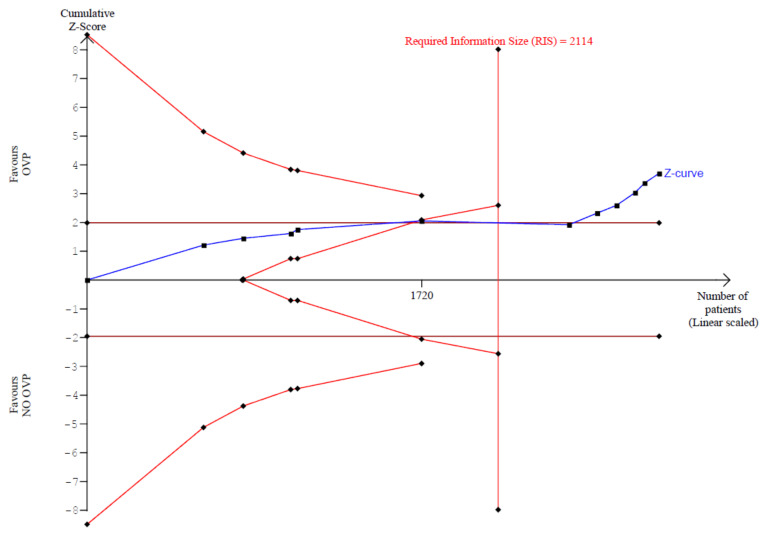
Trial sequential analysis of CDI onset comparing OVP with no active prophylaxis. Abbreviations: OVP: oral vancomycin prophylaxis.

**Figure 5 antibiotics-11-00183-f005:**
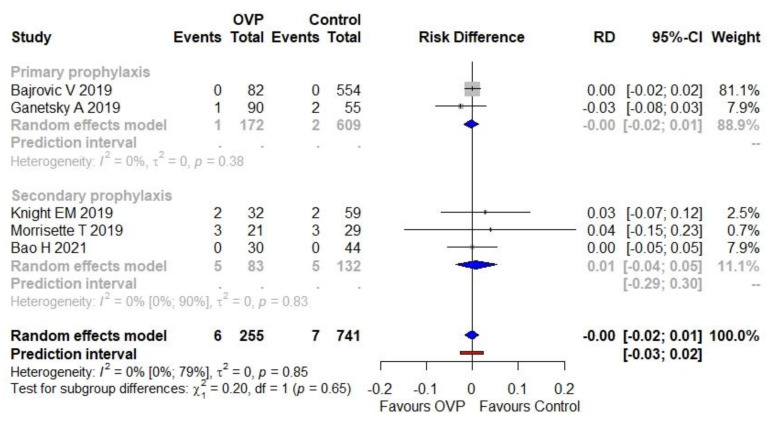
Meta-analysis regarding the risk difference between OVP group and comparators as the secondary outcome (VRE infections). Abbreviations: CDI, *Clostridioides difficile* infection; OVP, oral vancomycin prophylaxis; RD, risk difference; VRE, vancomycin-resistant *Enterococci*; 95%-CI, confidence intervals at 95%. Vertical line indicates the ‘no difference’ point between the two options. Squares represent adjusted odds ratios. Diamonds represent pooled risk difference for all studies. Horizontal lines represent 95% CI.

**Table 1 antibiotics-11-00183-t001:** Characteristics of included studies.

Author [Ref.]	Year of Publication	Study Design and Type of Prophylaxis	Study Period	N. Patients (OVP vs. Controls)	Characteristics of the Population	OVP Dose	OVP Duration	Event Definition	Time of Follow-Up	Incidence of CDI (OVP vs. Controls)	More Frequent Class of Antibiotic Used as SAT (OVP vs. Controls)
Carignan et al. [33]	2016	Retrospective cohort—Secondary	2003–2011	227 vs. 324	Adults receiving antibiotics within 90 days of initial/recurrent CDI/secondary prophylaxis	125 mg qid (84%)	7 days	diarrhea + toxin evidence or typical colitis	6 months	28% vs. 32%	NA (however, patients receiving second-generation cephalosporins more likely to experience rCDI)
Van Hise et al. [34]	2016	Retrospective cohort—Secondary	2010–2014	71 vs. 132	Adults with history of CDI, subsequently hospitalized and treated with systemic antimicrobial therapy/secondary prophylaxis	125 mg or 250 bid (59%)	14 days	diarrhea + NAAT	4 weeks	4.2% vs. 26.6% (*p* < 0.001)	Aminopenicillin, 49.3% vs. 47.7% (*p* = 0.88)
Papic et al. [35]	2018	Retrospective cohort—Primary	2015–2017	71 vs. 173	Elderly patients hospitalized for more than 72 h who received parenteral antibiotics for more than 24 h/primary prophylaxis	125 mg once daily	9 days	diarrhea + two-stage algorithm (GDH for screening and NAAT)	During index hospitalization	0% vs. 10.4% (*p* = 0.0022)	Piperacillin-tazobactam (53.5%) vs. any cephalosporin (43.4%)
Splinter et al. [36]	2018	Retrospective cohort—Secondary	2012–2015	11 vs.18	Adults renal transplanted patients with history of CDI/secondary prophylaxis	125 mg bid	19 days	NAAT	30 days	0% vs. 8 % (*p* = 0.54)	NA
Bajrovic et al. [37]	2019	Retrospective cohort—Primary	2007–2013	82 vs. 554	Adults receiving lung transplantation	125 mg bid (median)	14 days	diarrhea + NAAT	During index hospitalization	1.2% vs. 5.9%	Intravenous vancomycin, 100% vs. 69% (*p* < 0.01)
Caroff et al. [38]	2019	Retrospective cohort—Secondary	2009–2015	193 vs. 597	Adults given at least 1 dose of systemic antibiotic with history of CDI in previous 30–150 days/secondary prophylaxis	NA	2 days	toxin evidence or NAAT	90 days	9.8% vs. 9.4%	High-risk antibiotics according to study’s definition, 66% vs. 85% (*p* < 0.01)
Ganetsky et al. [39]	2019	Retrospective cohort—Primary *	2015–2016	90 vs. 55	Adults receiving allogenic hematopoietic cell transplantation	125 mg bid for the duration of stay	29 days	2/3-stage algorithm (GDH for screening, toxin detection or NAAT)	90 days	0% vs. 20% (*p* < 0.001)	Anti-Gram-negative antibiotics according to study’s definition, 76% vs. 71% (*p* = 0.54)
Knight et al. [40]	2019	Retrospective cohort—Secondary	2013–2015	32 vs. 59	Adults with history of CDI, subsequently hospitalized within 12 months and treated with systemic antimicrobials/secondary prophylaxis	125 or 250 (69%) mg qid	8.5 days	diarrhea + NAAT	12 months	6.3% vs. 27.8% (*p* = 0.011)	Penicillins vs. cephalosporins (in terms of sums of daily doses received)
Morrisette et al. [41]	2019	Retrospective cohort—Secondary	2014–2018	21 vs. 29	Hematological adults with and without HSCT treated for the initial episode of CDI first with planned oral vancomycin monotherapy and must have been receiving a BSA at time of CDI diagnosis and/or during the course of CDI treatment/prophylaxis	125 mg bid	7 days	diarrhea + NAAT	60 days	10% vs. 35% (*p* = 0.051)	Third/fourth generation cephalosporins, 95% vs. 93% (*p* > 0.99)
Johnson et al. [42]	2019	Randomized, prospective, open label—Primary	2018–2019	50 vs. 50	Adults admitted for more than 72 h, aged ≥60 years, hospitalized ≤30 days prior to the index hospitalization, and received systemic antibiotics during that prior hospitalization	125 mg once daily	12 days	diarrhea + NAAT	3 months post-discharge	0 vs. 12% (*p* = 0.03)	Cephalosporins (in terms of days of therapy, 173 vs. 171)
Bao et al. [43]	2021	Retrospective cohort—Secondary	2013–2019	30 vs. 44	Pediatric population, 50% affected by a malignancy	10 mg/kg (up to 125 mg per dose for non-severe CDI and 500 mg per dose for severe or fulminant CDI)	12 days	diarrhea + NAAT or typical colitis	8 weeks	3% vs. 25% (*p* = 0.02)	Third/fourth generation cephalosporins, 64% vs. 57% (*p* = 0.55)

Abbreviations: bid, bis in die; CDI, *Clostridioides difficile* infection; GDH, glutamate dehydrogenase; HSCT, hematopoietic stem cell transplantation; NA, not available; NAAT, nuclear acid amplification tests; OVP: oral vancomycin prophylaxis; qid, quarter in die; rCDI, recurrent *Clostridioides difficile* infection. * In this study the majority of patients underwent primary prophylaxis, but 15.6% of patients in the OVP arm had actually a previous episode of CDI compared with 9.1% of subjects in the comparator group (separate data were not available).

**Table 2 antibiotics-11-00183-t002:** Subgroup analysis of CDI occurrence under OVP versus no intervention in the overall population.

Variable	Included Studies, *n*	Sample Size, *n*	OR (95% CI)	I^2^	Test for Subgroup Difference, *p* Value
**Study place**					0.51
US	9	2095	0.15 (0.06–0.36)	59%	
Not US	2	845	0.31 (0.04–2.26)	0%	
**Study design**					0.99
Retrospective	10	2840	0.19 (0.09–0.44)	59.4%	
Prospective	1	100	0.07 (0.00–1.24)	-	
**Study population**					0.30
Adult hospitalized	6	1999	0.27 (0.10–0.78)	64.1%	
SOT	2	672	0.16 (0.02–1.21)	0%	
Hematological	2	195	0.03 (0.00–0.23)	0%	
Pediatric	1	74	0.10 (0.01–0.85)	-	
**OVP dose ***					0.11
125 mg od	2	344	0.06 (0.01–0.48)	0%	
125 bid	5	951	0.11 (0.04–0.32)	0%	
Other (variable/mixed dosages)	4	1645	0.43 (0.15–1.23)	78.5%	
**Timing of follow-up**					<0.01
28/30-day	2	239	0.12 (0.03–0.39)	0%	
90-day	3	1461	0.82 (0.61–1.11)	0%	
In-hospital	3	1025	0.03 (0.00–0.22)	0%	
Other	3	207	0.12 (0.04–0.36)	0%	
**Mean duration of OVP**					0.01
**(Compared with SAT)**					
Longer	7	1244	0.08 (0.03–0.18)	0%	
Shorter	4	1696	0.44 (0.16–1.23)	40%	

* The analysis was not run under a GLMM as this model could not be fit, but rather through MH method without continuity correction. Abbreviations: bis: bis in die; CDI: *Clostridioides difficile* infection; MH, Mantel–Haenszel; od, once a day; OR, odds ratio; OVP, oral vancomycin prophylaxis; SAT, systemic antibiotic therapy; SOT, solid organ transplantation; US, United States; 95%-CI, confidence intervals at 95%.

## Data Availability

The datasets generated during the current meta-analysis are available from the corresponding author upon reasonable request. All data analyzed for meta-analysis are included in the corresponding published articles, as reported in Table 1.

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
