# Peer review of "Oral Vancomycin Prophylaxis for Primary and Secondary Prevention of *Clostridioides difficile* Infection in Patients Treated with Systemic Antibiotic Therapy: A Systematic Review, Meta-Analysis and Trial Sequential Analysis"

_antibiotics, 2022, doi:10.3390/antibiotics11020183_

Round 1

Reviewer 1 Report

This paper by Maraolo et al. aims to evaluate the role of oral vancomycin prophylaxis for primary and secondary prevention of CDI in subjects undergoing SAT. This is a high quality article that introduces and discusses the theme thoroughly. Well done. This paper adds to the current knowledge on the use of OVP in SAT.

Below I leave some comments that the authors may answer by adding some more information to the original article:

1 - It is known that stopping PPIs on patients undergoing SAT decreases the likelihood of an established CDI (more than stopping H2 inhibitors). Is there any information on the risks/benefits of this VS OVP? Even if not, authors can address this in the paper, allowing some thought for the urgency of decreasing the number of globally prescribed PPIs and this impact this has overall.

2 - Where is this line that measures risk/benefit of OVP and what are the authors' suggestions (taking into account the well addressed limitations of the study)? In other words, what is the recommendation of this paper regarding OVP and its risks: use it but to be stricter in the definition of risk population? Use it in the lowest dose? Choose OVP for selected SAT antibiotics? This papers' discussion and conclusions are brilliant, but I believe the text needs more assertiveness by matching the current knowledge with this papers' results, at the end. Although hypothetical, conclusions like these can guide further studies.

3 - Oral vancomycin (capsules) is expensive compared to (unlicensed) oral liquid vancomycin taken from IV bottles. This is based in the current clinical guidances in the UK. For patients on SAT outside hospitals, capsules need to be taken and this can quite expensive for national health services. What are the thoughts of the authors regarding this? Is OVP only advised in a hospital setting where liquid vancomycin can used orally?

Overall, very good paper with new key information regarding OVP in SAT.

Author Response

We thank the reviewer for her/his useful comments.

We enriched discussion and conclusion according to her/his observations.

1) We added lines 650-656 and ref. 68 to address the role of PPI as important risk factor for CDI, endorsing a deprescribing approach.

2) We added lines 668-671 to define our proposal to use OVP.

3) We added line 658-662 and ref. 69 to address the issue of potentially high costs of generic vancomycin capsules.

Reviewer 2 Report

It is a well designed work and can be accepted for publication and its conclusion is helpful in its field.

Author Response

We thank the referee for his/her approval of the manuscript.

Reviewer 3 Report

The manuscript is well written, well-structured and adequately cited.

I have no major comments regarding this manuscript which is of publishable quality.

Author Response

We thank the referee for her/his approval of the manuscript.